# Decidualized Endometrial Stromal Cells Promote Mitochondrial Beta-Oxidation to Produce the Octanoic Acid Required for Implantation

**DOI:** 10.3390/biom14081014

**Published:** 2024-08-16

**Authors:** Yumi Mizuno, Shunsuke Tamaru, Hideno Tochigi, Tomomi Sato, Miyuko Kishi, Akira Ohtake, Osamu Ishihara, Takeshi Kajihara

**Affiliations:** 1Department of Obstetrics and Gynecology, Faculty of Medicine, Saitama Medical University, Iruma, Moroyama 350-0495, Saitama, Japan; yumimi@saitama-med.ac.jp (Y.M.); stamaru@saitama-med.ac.jp (S.T.); hideno@saitama-med.ac.jp (H.T.); tosato@saitama-med.ac.jp (T.S.); wkishi@saitama-med.ac.jp (M.K.); ishihara.osamu@eiyo.ac.jp (O.I.); 2Biomedical Research Center, Faculty of Medicine, Saitama Medical University, Iruma, Moroyama 350-0495, Saitama, Japan; 3Department of Anatomy, Saitama Medical University, Iruma, Moroyama 350-0495, Saitama, Japan; 4Department of Clinical Genomics, Faculty of Medicine, Saitama Medical University, Iruma, Moroyama 350-0495, Saitama, Japan; akira_oh@saitama-med.ac.jp; 5Department of Pediatrics, Faculty of Medicine, Saitama Medical University, Iruma, Moroyama 350-0495, Saitama, Japan; 6Nutrition Clinic, Kagawa Nutrition University, Toshima 170-8481, Tokyo, Japan

**Keywords:** decidualization, octanoic acid, mitochondrial beta-oxidation, embryo implantation

## Abstract

Decidualization denotes the morphological and biological differentiating process of human endometrial stromal cells (HESCs). Fatty acid pathways are critical for endometrial decidualization. However, the participation of fatty acids as an energy source and their role in endometrial decidualization have received little attention. To identify fatty acids and clarify their role in decidualization, we comprehensively evaluated free fatty acid profiles using liquid chromatography/Fourier transform mass spectrometry (LC/FT-MS). LC/FT-MS analysis detected 26 kinds of fatty acids in the culture medium of decidualized or un-decidualized HESCs. Only the production of octanoic acid, which is an essential energy source for embryonic development, was increased upon decidualization. The expressions of genes related to octanoic acid metabolism including ACADL, ACADM, and ACADS; genes encoding proteins catalyzing the first step of mitochondrial fatty acid beta-oxidation; and ACSL5 and ACSM5; genes encoding fatty acid synthesis proteins were significantly altered upon decidualization. These results suggest that decidualization promotes lipid metabolism, implying that decidualized HESCs require energy metabolism of the mitochondria in embryo implantation.

## 1. Introduction

Pregnancy requires the coordination of three interdependent processes, including embryo development, endometrial decidualization, and placenta formation. The interaction between invasive trophoblasts and the maternal uterine decidua is critical for successful implantation and pregnancy [1]. Decidualization denotes the morphological and biological differentiating process of human endometrial stromal cells (HESCs), initiated in the middle-to-late secretory phase of the cycle. Impaired decidualization causes a spectrum of pregnancy complication, including implantation failure, recurrent miscarriage, and preeclampsia [2]. Decidualization is characterized by the transformation of endometrial stromal cells into epithelioid-like decidual cells, a process that is further characterized by the influx of specialized immune cells and macrophages with intense vascular remodeling [3,4]. Previously, we demonstrated that decidualized HESCs caused ultrastructural morphological alternations, including expanded endoplasmic reticulum, and increased numbers of mitochondria and lipid droplets [5].

The initial contact of the implanting blastocyst with the lumen epithelium of the endometrial lining occurs during the steps of apposition and attachment at 6–7 days following conception. As soon as the epithelium is breached, the interface emerges between trophoblasts and decidual cells. Upon breaching of the luminal endometrial epithelium, the implantation embryo is rapidly surrounded and encapsulated by migrating decidualizing endometrial stromal cells. Thus, decidual–trophoblast crosstalk is essential for successful pregnancy. Gerllersen et al. [6] showed that when cultured on decidualized HESCs, trophoblast spheroids expand to a greater extent than un-differentiated HESCs. Decidualized HESCs are highly secretory cells, suggesting that they create the microenvironment and provide the nutrients that enable the conceptus to thrive [7].

Several studies have revealed that glucose metabolism plays a significant role in endometrial decidualization [8,9,10]. In addition to glucose metabolism, several cell types depend on fatty acids as an energy source. Lipids play a crucial role in the physical properties and biological functions of membranes as well as providing the cell a source of nutrients [11]. Lipid metabolism is involved in the luteal phase endometrial functionality [12,13]. Lipid metabolism activation is a well-represented process in the secretory phase of the human endometrium [14]. Furthermore, fatty acid beta-oxidation pathways are critical for endometrial decidualization [15]. However, the participation of fatty acids as an energy source and their roles in endometrial decidualization have received little attention. Our understanding of the expression, regulation, and function of fatty acids in differentiating HESCs is limited. 

In previous studies, Yamada et al. [16] suggested that medium-chain fatty acid is an alternative energy source in mouse preimplantation development. Although it was found that medium-chain fatty acids are necessary for embryo development, there were still some unknowns, such as where they are supplied from and whether they play a role in decidualization and implantation. Our previous study demonstrated that decidualized HESCs had an increased number of lipid droplets, suggesting that they may be a potential source of medium-chain fatty acids [5]. Therefore, to clarify the relationship between medium-chain fatty acids and endometrial decidualization or embryo implantation, we created a free fatty acid profile using liquid chromatography/Fourier transform mass spectrometry (LC/FT-MS) and analyzed the expression levels of related genes and invasion.

## 2. Materials and Methods

### 2.1. Tissue Collection and Primary Culture of HESCs

This study conducted following ethical approval from the Institutional Review Board of Saitama Medical University Hospital (IRB code no.16001, 11017). Human endometrial tissues were collected from four women, who underwent hysterectomy for myoma uteri at Saitama Medical University Hospital. Subjects were not on hormonal treatment at least 6 months prior to the procedure. All patients gave written informed consent to the study before the operation and the study protocol was approved by the local ethical committee of Saitama Medical University Hospital. For primary culture, HESCs were isolated from the uterus following hysterectomy as previously described [17,18,19]. The isolated HESCs were cultured in DMEM/F-12 (Thermo Fisher Scientific, Waltham, MA, USA) containing 10% dextran-coated charcoal (DCC)-treated Fetal bovine serum; FBS (Nichirei Biosciences, Tokyo, Japan) with 1% antibiotic–antimycotic solution (Thermo Fisher Scientific, Waltham, MA, USA) and 2 mM L-Glutamine (Thermo Fisher Scientific, Waltham, MA, USA).

### 2.2. Decidualization of HESCs

HESCs were decidualized by 0.5 mM Adenosine 3′,5′-cyclic Monophosphate, 8-Bromo-, Sodium Salt (8-br-cAMP B7880; Merck, Sigma-Aldrich, Darmstadt, Germany) and 10−6 M medroxyprogesterone 17-acetate (MPA M1629; Merck, Sigma-Aldrich, Darmstadt, Germany) in DMEM/F-12 containing 2% DCC-treated FBS. All experiments were conducted before the third passage of the cultures.

### 2.3. LC/FT-MS

LC/FT-MS analysis of fatty acids was outsourced to the Chemicals Evaluation and Research Institute (CERI, Tokyo, Japan) and performed as previously reported [20]. LC detection was performed using UFLC XR (Shimadzu, Kyoto, Japan). Chromatographic separation was performed on a one-column 2 ODS column (2.1 × 150 mm; particle size, 2 μm) (CERI, Tokyo, Japan). MS was performed using LTQ Orbitrap XL (Thermo Fisher Scientific, Waltham, MA, USA). MS detection was performed in a negative scan mode at a resolution of 30,000 and a range of *m/z* 140–600.

### 2.4. Beta-Oxidation Assay

Beta-oxidation activity was measured using Radioisotope (RI)-labeled substrate as previously reported [21]. Mitochondrial fatty acid oxidation activity was measured with RI-labeled fatty acids following [1–14C]-labeled capric acids (C10:0) (PerkinElmer, Shelton, CT, USA) or [1–14C]-labeled palmitic acid (C16:0) (PerkinElmer, Shelton, CT, USA). Briefly, mitochondrial fatty acid oxidation activity was measured using decidualized or un-decidualized HESCs. HESCs were dissociated with trypsin, centrifuged to remove the medium, and then resuspended in a reaction solution containing RI-labeled fatty acids and incubated at 37 °C for 1 h. To extract the 14C-labeled metabolites after the beta oxidation reaction, heated at 60 °C with 1/5 volume of 1N KOH and acidified in 6% perchloric acid for 1 h on ice. After centrifugation at 7700× *g* for 10 min, chloroform; methanol (2:1) extraction was performed. 14C-labeled metabolites were analyzed in a liquid scintillation counter LS 6500 (Beckman Coulter, Brea, CA, USA).

### 2.5. RNA Extraction and Expression

The total RNA was extracted from decidualized or un-decidualized primary cultures of HESCs using miRNeasy mini prep kit (Qiagen, Hilden, Germany). Reverse transcription was performed using BioScript Reverse Transcriptase (Bioline, Memphis, TN, USA). Each gene expression was analyzed by the PikoReal 96 Real time PCR system (Thermo Fisher Scientific, Waltham, MA, USA) using PowerUp SYBR Green Master Mix (Thermo Fisher Scientific, Waltham, MA, USA).

### 2.6. Invasion Assay

Invasion assay was performed using extravillous trophoblast cells (HTR8/SVneo) cultured within a Boyden chamber with 8-mm pores (Chemotaxicell; Kurabo, Osaka, Japan). An immortalised first trimester EVT cell line (HTR-8/SVneo cells) was used for invasion assay. HTR-8/SVneo cell lines were kindly provided by Dr. Charles H. Graham (Queen’s University, Kingston, ON, Canada) and Dr. Eiko Yamamoto (Nagoya University Graduate School of Medicine, Nagoya, Japan) [22]. HTR8/SVneo cells (1.0 × 10^5^ cells suspended in DMEM/F-12 containing 10% FBS) were added into a chemotaxis–cell chamber placed on a 24-well plate. Octanoic acid (200 ng/mL) was added in a 1 mL culture medium (DMEM/F-12 containing 10% FBS) per well and incubated with HTR8/SVneo cells in the chamber for 2 days. HTR8/SVneo cells invading toward the outer surface of the membrane through pores were fixed and stained with the Diff-Quik stain (Sysmex, Hyogo, Japan). The HTR8/SVneo cells on the outer surface of the membrane were counted, and the average number of invaded cells was calculated from five fields under 100× objectives per chamber. Quantification of invaded HTR-8/SVneo cells was performed by counting the number of invaded cells per pixel in micrographs and calculating the average. Statistical analysis was performed using the Mann–Whitney U test.

## 3. Results

### 3.1. Effects of Decidualizing Stimuli Using 8-br-cAMP and MPA in HESCs

Primary HESCs upon decidualization by 8-br-cAMP and MPA showed a typical morphology of decidualized cells (Figure 1a). IGFBP1 and PRL mRNA expressions were significantly increased by decidualization (Figure 1b).

### 3.2. Promoted Production of Octanoic Acid in the Culture Medium of Decidualized HESCs Revealed by Free Fatty Acid Profile Analysis

Free fatty acid profiles were comprehensively evaluated in the culture medium of decidualized and un-decidualized HESCs with or without 8-br-cAMP and MPA, respectively, using LC/FT-MS. A total of 66 kinds of medium-chain fatty acids and long chain fatty acids were assessed. Thirty-two kinds of medium-chain fatty acid and long chain fatty acids were detected using LC/FT-MS in a 3-day culture medium of un-decidualized or decidualized HESCs. Only octanoic acid, which is one of the medium-chain fatty acid s, was increased in the culture medium of decidualized HESCs compared with un-decidualized HESCs (Figure 2). In contrast, the concentration of the other 25 kinds of medium-chain fatty acid and long-chain fatty acids was not increased upon decidualization (Figure 2). Unsaturated fatty acids were also not increased but rather decreased upon decidualization. Therefore, we subsequently focused on octanoic acid metabolism in decidualized HESCs.

### 3.3. Fatty Acid Beta-Oxidation Activity Increased in Decidualized HESCs

We evaluated the fatty acid beta-oxidation activity using straight-chain fatty acids with the radiolabel at position C-1. As ^14^C-labeled capric acids (C10:0) medium-chain fatty acid was oxidized via the beta-oxidation process to generate acetyl-CoA; the fatty acid oxidation rate was measured by the incomplete oxidation of capric acids to octanoic acids (C8:0). The medium-chain fatty oxidation rate increased upon HESC decidualization by 8-br-cAMP and MPA (Figure 3a). Furthermore, the fatty acid beta-oxidation activity using ^14^C-labeled palmitic acids (C16:0) was increased upon HESC decidualization (Figure 3b).

### 3.4. Expression of Genes Related to Fatty Acid Metabolism in Decidualized HESCs

We evaluated the expression profiles of genes associated with synthesis and decomposition of long-chain fatty acids, medium-chain fatty acids, and short chain fatty acids. The beta-oxidation process is composed of a repeated sequence of several reactions, including catalyzation by acyl-CoA dehydrogenases such as ACADL, ACADM, and ACADS; these were evaluated for the expression changes (Figure 4). Upon HESC decidualization by 8-br-cAMP and MPA, only the ACADM mRNA expression was downregulated (Figure 4). ACADM is one of the acyl-CoA dehydrogenases with the substrate specificity ranging from C12 to C4 medium-chain fatty acids; ACADM promotes octanoic acid decomposition [23]. Upon HESC decidualization, the gene expression of ACADS, an acyl-CoA dehydrogenase short chain, was upregulated in contrast to that of ACADM (Figure 4). The gene expression of ACADL, an acyl-CoA dehydrogenase long chain, showed no significant difference between decidualized and un-decidualized HESCs (Figure 4). A gene encoding CPT1B, Carnitine O-Palmitoyltransferase 1B, which is one of the rate-controlling enzymes CPT1 in long-chain fatty acid beta-oxidation pathway, was downregulated (Figure 4). The process of fatty acid activation by CoA to produce an acyl-CoA is composed of several acyl-CoA synthetases, such as ACSLs and ACSMs. ACSM5 catalyzes fatty acid activation, the first step in fatty acid metabolism that induces octanoic acid production. Upon HESC decidualization by 8-br-cAMP and MPA, the ACSM5 mRNA expression was upregulated (Figure 4).

### 3.5. Tendency to Promote Invasion by Octanoic Acid in a Trophoblast Cell Line

To explore the functional role of octanoic acid in implantation, we investigated the effect on invasion in an extravillous trophoblast cell line, HTR-8/SVneo. Using a transwell cell invasion assay, we revealed that octanoic acid showed a tendency to promote HTR-8/SVneo cell invasion; however, it was not significantly different, suggesting a positive role of octanoic acid in trophoblast invasion (Figure 5).

## 4. Discussion

We here demonstrated that decidualized processes (Figure 1) promoted the mitochondrial lipid metabolism in HESCs. Moreover, the production of octanoic acid, which is one of the medium-chain fatty acids, was only increased (Figure 2).

Octanoic acid is a protein stabilizer generally added to the preparations of human serum albumin used for culture of human embryos in clinical assisted reproductive technology. However, octanoic acid at high concentrations in the culture medium of embryos causes disruptions in mitochondrial bioenergetics, which reduces intracellular pH to induce oxidative damage in peripheral tissues, and finally inhibits embryonic development [24]. Additionally, Fredrickson et al. [25] reported that octanoic acid could produce long-term negative effects on embryonic and fetal development in a murine model. Conversely, Yamada et al. [16] revealed that octanoic acid is an energy source throughout the preimplantation development in mice, as mice embryos did not survive in the culture medium that lacked fatty acids, pyruvate, and glucose, whereas octanoic acid supplementation rescues the embryonic development. In the present study, octanoic acid concentration increased in the culture medium of decidualized HESCs (Figure 2), and the metabolism was regulated by an increase in synthesis by higher ACSM5 expression and a decrease in decomposition by lower ACADM expression (Figure 4). These results suggest that increased octanoic acid production changes in the expression of ACSM5 and ACADM in decidualized HESCs would play important roles in modulating the energy metabolism during implantation period.

The expression of genes encoding beta-oxidation enzymes including ACADL, ACADM, and ACADS that catalyze acyl-CoA dehydrogenases suggests interesting possibilities from the point of view of fatty acid metabolism and energy source for preimplantation embryos. Our results showed that only the ACADM gene expression was downregulated, whereas those of ACADL and ACADS were not (Figure 4). These results indicate that medium-chain fatty acids are regulated to reduce their intracellular consumption by suppressing the ACADM expression. Moreover, medium-chain fatty acids especially octanoic acids may be secreted from decidualized HESCs to play a biological role in the reproductive system. The expression of genes related to octanoic acid metabolism such as ACADM and ACSM5 was altered in the culture medium of decidualized HESCs in comparison with un-decidualized HESCs. Octanoic acid directly enters the mitochondrial matrix [23] and would be used as an alternative energy source throughout preimplantation processes, including trophoblast invasion and placental development. Therefore, mitochondrial fatty acid beta-oxidation, a catabolic process of fatty acid breakdown, showed a large increase of 2.4-fold in the oxidation rate of medium-chain fatty acids (C10:0) due to decidualization of HESCs (Figure 3a). In contrast, beta-oxidation of long-chain fatty acids (C16:0) increased by approximately 1.3-fold, although there was a significant difference (Figure 3b). 

When there is sufficient long-chain fatty acid (C16:0) as a substrate, beta-oxidation activity of fatty acids increases with decidualization (Figure 3b). However, CPT1B is decreased by decidualization (Figure 4), and the gene expression level of ACSL5, which synthesizes long-chain fatty acids, is also decreased (Figure 4). These results suggest that the uptake of long-chain fatty acids into mitochondria is regulated in two steps in decidualized HESCs: the decrease in synthesis of long-chain fatty acids in the cytoplasm and the uptake of long-chain fatty acids into mitochondria by CPT1. As a result, only the mild increase of long-chain fatty acids beta-oxidation activity in decidualized HESCs which is thought to be due to the effect of the decrease in CPT1 (Figure 3 and Figure 4).

Beta-oxidation is a process by which fatty acid molecules are disrupted in the mitochondria to generate acetyl-CoA, which enters the citric acid cycle, and NADH and FADH2, which are co-enzymes used in the electron transport chain to generate the energy. Tsai et al. [15] showed that CPT1, the rate-controlling enzyme of the long-chain fatty acid beta-oxidation pathway, is significant for human and murine ESC decidualization. When the CPT1 activity was reduced, the expression of decidualized markers was inhibited in decidualized ESCs [15]. To date, this is the only study describing the significance of the beta-oxidation pathway in HESC decidualization. Our study demonstrated that the beta-oxidation activity of medium-chain fatty acids significantly increased upon HESC decidualization (Figure 3a), indicating that the beta-oxidation activity of medium-chain fatty acids is increased owing to the production from capric acid (C10:0) to octanoic acid (C8:0). Although two phenomena including fatty acid beta-oxidation activity induction and ACADM gene expression reduction simultaneously occurred, both “disruption” and “production” pathways produce octanoic acids in a coordinated manner during decidualization. Furthermore, even when the beta-oxidation activity of long-chain fatty acids is increased, this also implies that the activation of beta-oxidation promotes the use of fatty acids as a source of energy substrate.

Decidual–trophoblast dialogue is critical for a successful pregnancy. The decidua is believed to regulate trophoblast invasion and placental development by the secretion of locally produced factors, including cytokines, hormones, metabolites, and dietary compounds [26]. Several lines of evidence suggest that fatty acids play a significant role in the embryo implantation process [27,28,29]. The maternal supply of short chain fatty acids and midium chain fatty acids during early pregnancy potentiates uterine phospholipid metabolism, thereby leading to improved embryo survival [30]. In prostate cancer cells, octanoic acid promotes invasion [31]. Therefore, we hypothesized that octanoic acid secreted by decidualized HESCs promotes invasion capacity of trophoblast cells. We employed invasion assay using by HTR-8/SVneo trophoblast cells. Our observation implied that octanoic acid tended to promote invasion in HTR-8/SVneo trophoblast cells. This finding suggests that octanoic acid secreted by decidualized HESCs can act to facilitate trophoblast invasion processes. However, to clarify the precise role of octanoic acid derived from decidualized HESCs in implantation processes, further studies are needed.

Large efforts have been invested to identify the best biomarker that characterizes a receptive endometrium. In the last decade, different “omics” technologies including genomics, proteomics, and lipidomics were employed and revealed large number of candidates of biomarkers for a receptive endometrium. Vilella et al. [32] reported that lipidomics is an emerging tool for predicting endometrial receptivity. Our study provides a possibility that octanoic acid is a new potential biomarker for predictive endometrial receptivity. However, further studies are needed for octanoic acid as clinical applications of a biomarker to predict endometrial receptivity. In addition, this study revealed that many types of long-chain fatty acids and unsaturated fatty acids in the culture medium were reduced by decidualization, indicating that endometrial cells require a large of fatty acids for the decidualization. In the future, investigating how the presence of various fatty acids affects endometrial decidualization may be beneficial for the development.

## 5. Conclusions

This study shows, for the first time, that HESCs decidualization promotes octanoic acid production that modulates by cytosol fatty acid production and mitochondrial lipid metabolisms. These changes suggest the requirement of fatty acid metabolism with mitochondrial role in HESCs decidualization and embryo implantation. 

## Figures and Tables

**Figure 1 biomolecules-14-01014-f001:**
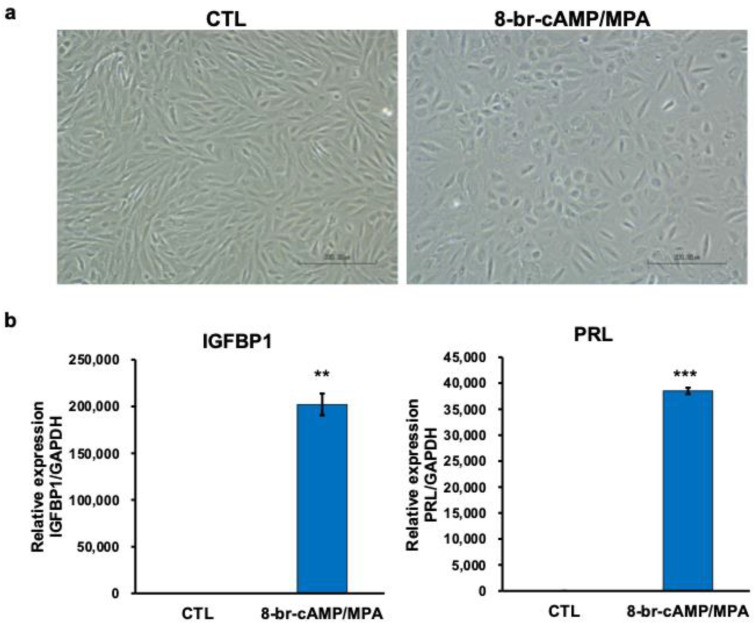
Effects of decidualizing stimuli using 8-br-cAMP and MPA in HESCs. (**a**) Morphological changes in HESCs. Undifferentiated primary HESCs exhibit a fibroblastic spindle-shaped morphology (CTL). Primary HESCs, upon decidualization by 8-br-cAMP and MPA, show abundant cytoplasm and larger nuclei, which are the typical morphology of decidualized cells. (**b**) Primary HESCs are stimulated in the absence (CTL) or presence of 8-br-cAMP and MPA. IGFBP1 and PRL mRNA expressions are measured using RT-qPCR. Results were normalized to GAPDH and compaired with CTL was set as 1. Statistical analysis performed by two-tailed Student’s *t*-tests. Data are shown as the mean ± SEM (standard error of the mean, *n* = 3). ** *p* < 0.01, *** *p* < 0.001.

**Figure 2 biomolecules-14-01014-f002:**
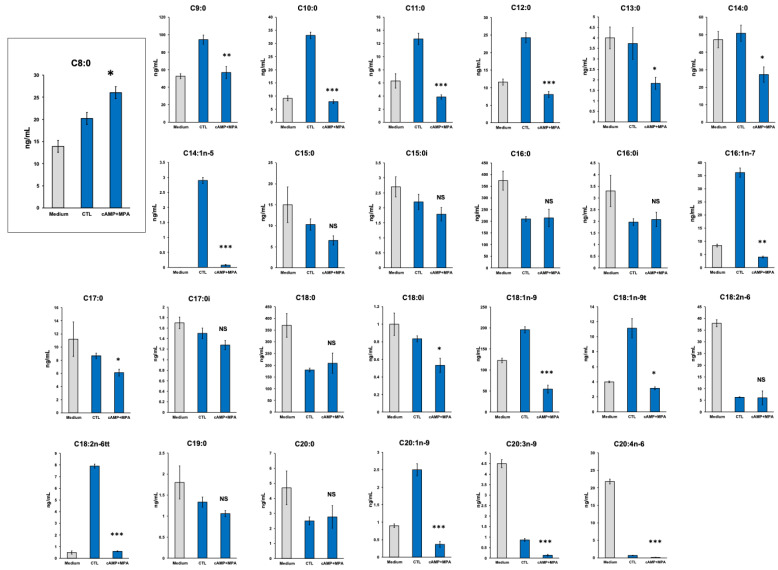
Evaluation of 26 kinds of medium-chain fatty acids and long-chain fatty acids in the culture medium of decidualized HESCs analyzed using LC/FT-MS. All kinds of fatty acids but octanoic acid did not increase upon decidualization in the culture medium. Statistical analysis of un-decidualized (CTL) and decidualized (cAMP+MPA) are performed by two-tailed Student’s *t*-tests. Data are shown as the mean ± SEM (standard error of the mean, *n* = 3–4). * *p* < 0.05, ** *p* < 0.01, *** *p* < 0.001. NS; No significant difference. “Medium” refers to the culture medium before use.

**Figure 3 biomolecules-14-01014-f003:**
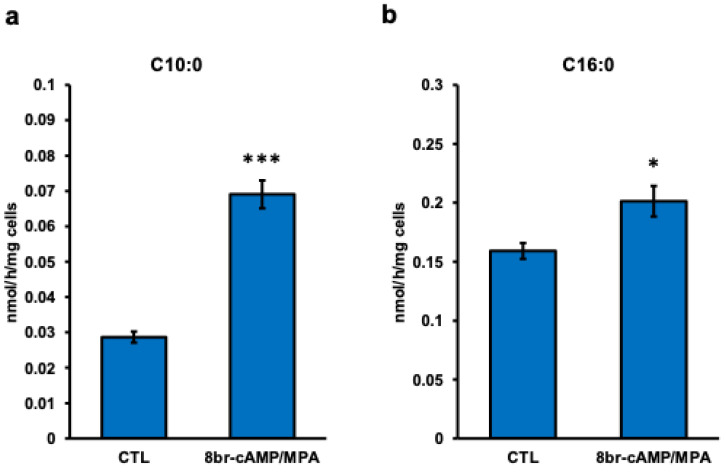
Evaluation of the fatty acid beta-oxidation activity by palmitate with the radiolabel at position C-1. Primary HESCs are stimulated in the absence (CTL) or presence of 8-br-cAMP and MPA, and the incomplete oxidation of [1–14C]-labeled capric acids (C10:0) (**a**) or [1–14C]-labeled palmitic acid (C16:0) (**b**) is measured as the fatty acid oxidation rate. Statistical analysis performed by two-tailed Student’s *t*-tests. Data are shown as the mean ± SEM (standard error of the mean, *n* = 6). * *p* < 0.05, *** *p* < 0.001.

**Figure 4 biomolecules-14-01014-f004:**
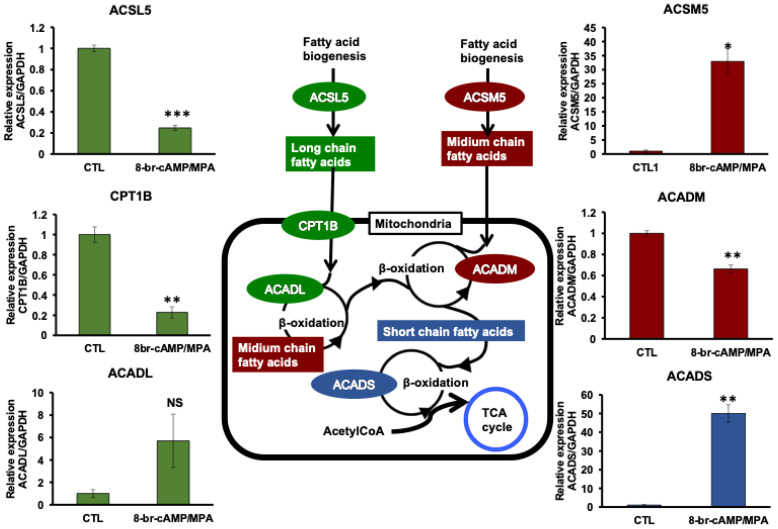
Expression profiles of the genes associated with the beta-oxidation process. Primary HESCs are stimulated in the absence (CTL) or presence of 8-br-cAMP and MPA. The expression of each gene mRNA are measured using RT-qPCR and normalized to GAPDH. The diagram in the middle shows the metabolic pathways of fatty acids. Expressions of long chain (green), medium-chain (red), and short chain (blue) fatty acid metabolism-related genes are analyzed using RT-PCR. Statistical analysis performed by two-tailed Student’s *t*-tests. Data are shown as the mean ± SEM (standard error of the mean, *n* = 3-6). * *p* < 0.05, ** *p* < 0.01, *** *p* < 0.001, NS; No significant difference.

**Figure 5 biomolecules-14-01014-f005:**
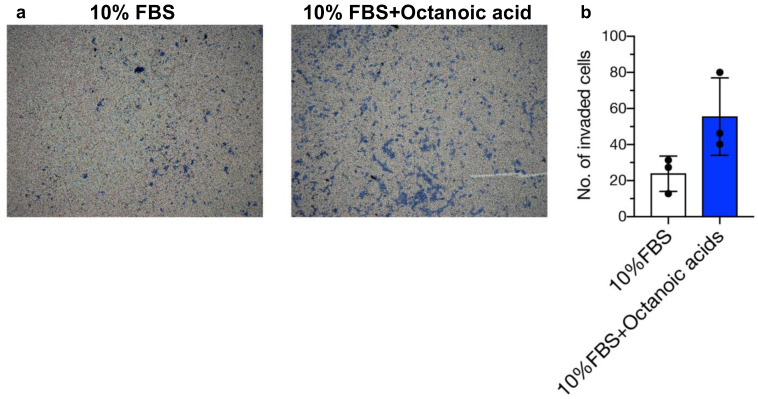
Effects of octanoic acid addition in culture medium for HTR-8/SVneo cell invasion. (**a**) HTR-8/SVneo cells invaded by incubation with control (10% FBS) or octanoic acid (200 ng/mL in 10% FBS). (**b**) Quantification of the average number of invaded HTR-8/SVneo cells (mean ± SEM, *n* = 3, *p* = 0.1; Mann–Whitney U test). Each experiment is performed in triplicate (*n* = 3). The black dots represent the average of five fields for each experiment.

## Data Availability

The datasets generated and/or analyzed during the current study are available from the corresponding author on reasonable request.

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
