# Peer review of "Decidualized Endometrial Stromal Cells Promote Mitochondrial Beta-Oxidation to Produce the Octanoic Acid Required for Implantation"

_biomolecules, 2024, doi:10.3390/biom14081014_

Round 1

Reviewer 1 Report

Comments and Suggestions for Authors

Manuscript entitled "Decidualized endometrial stromal cells promote mitochondrial beta-oxidation to produce the octanoic acid required for implantation" by Mizuno et al., shows regulation of fatty acid metabolism associated with decidualization of endometrial stromal cells. The result show increased concentration of octanoic acid in decidualized cells and its  possible involvement in trophoblast cell invasion. The data regarding role of octanoic acid in embryo development are conflicting, however this report shows a tight regulation of octanoic acid  is required for embryo implantation.

Minor comments:

The methods related to beta-oxidation assay needs to be elaborated.

Please justify decrease in CPT1 with decidualization in present study, a rate limiting enzyme in beta oxidation of fatty acids. 

Though the presented in vitro studies show fatty acid metabolism to produce octanoic acid during decidualization but results are not sufficient to suggest it as a receptivity markers.

Line 19: remove employed

Line 69: Tissue collection

Line 76: For primary culture, HESCs were isolated

Comments on the Quality of English Language

English Language is fine, minor editing required

Author Response

The methods related to beta-oxidation assay needs to be elaborated.

Authors’ response: The method for the beta-oxidation assay is described in detail.

Please justify decrease in CPT1 with decidualization in present study, a rate limiting enzyme in beta oxidation of fatty acids. 

Authors’ response: Figure 3b has shown that beta-oxidation activity of fatty acids increases significantly with decidualization when there is sufficient long-chain fatty acid (C16:0) as a substrate. However, CPT1B is decreased by decidualization (Figure 4), and the gene expression level of ACSL5, which synthesizes long-chain fatty acids, is also decreased (Figure 4). These results suggest that the uptake of long-chain fatty acids into mitochondria is regulated in two steps in decidualized HESCs: the decrease in synthesis of long-chain fatty acids in the cytoplasm and the uptake of long-chain fatty acids into mitochondria by CPT1. As a result, the beta-oxidation activity of decidualized HESCs was approximately 2.41 times higher for medium-chain fatty acids, while it was approximately 1.26 times higher for long-chain fatty acids. These sentences were described in the Discussion.

Though the presented in vitro studies show fatty acid metabolism to produce octanoic acid during decidualization but results are not sufficient to suggest it as a receptivity markers.

Authors’ response: Thank you for this great suggestion. According to your comment, we edited discussion part as below

Our study provides a possibility that octanoic acid is a new potential biomarker for predictive endometrial receptivity. However, further studies are needed about octanoic acid as clinical applications of a biomarker to predict endometrial receptivity.

Line 19: remove employed

Authors’ response: We removed “employed”

Line 69: Tissue collection

Authors’ response: We correct.

Line 76: For primary culture, HESCs were isolated

Authors’ response: We correct “Primary culture” to “For primary culture,”

Reviewer 2 Report

Comments and Suggestions for Authors

In this study, the focus was on human endometrial stromal cells decidualization promotes octanoic acid production that modulates mitochondrial lipid metabolisms, which required some modifications.

1. In the introduction, the authors' description of the current state of research on the role of fatty acids pathways in endometrial decidualization is one-sided and does not correlate well with the conclusions of the paper.

2. The groupings in the statistical graphs could perhaps be designed with different colors for differentiation.

3. In Figure 5, the authors should have accounted for the method used to quantify the number of cells.

4. In the Discussion section, the authors only discuss Figure 4, while the other images are not discussed, and this section needs to be revised .

5. The authors have only performed cellular experiments, and it would be better to perform in vivo experiments if available.

Author Response

In this study, the focus was on human endometrial stromal cells decidualization promotes octanoic acid production that modulates mitochondrial lipid metabolisms, which required some modifications.

Authors’ response: In the statistical analysis in Figure 2, the graphs are color-coded differently from the culture medium graphs in order to compare only the control and decidualized groups.3.

  1. In the introduction, the authors' description of the current state of research on the role of fatty acids pathways in endometrial decidualization is one-sided and does not correlate well with the conclusions of the paper.

Authors’ response: Thank you for this great suggestion. According to your comment, we edited discussion part as below

We here demonstrated that decidualized processes promoted the mitochondrial lipid metabolism in HESCs. Moreover, the production of octanoic acid, which is one of the MCFAs, was increased, and the expression of genes related to octanoic acid metabolism such as ACADM and ACSM5 was altered in the culture medium of decidualized HESCs in comparison with non-decidualized HESCs. Octanoic acid directly enters the mito-chondrial matrix [22] and would be used as an alternative energy source throughout preimplantation processes, including trophoblast invasion and placental development. Therefore, the fatty acid beta-oxidation in the mitochondria, which is the catabolic process of fatty acid decomposition, showed that the MCFA and LCFA oxidation rates were in-creased upon HESC decidualization (Figure 4).

  1. The groupings in the statistical graphs could perhaps be designed with different colors for differentiation.

Authors’ response: In the statistical analysis in Figure 2, the graphs are color-coded differently from the culture medium graphs in order to compare only the control and decidualized groups.3.

  1. In Figure 5, the authors should have accounted for the method used to quantify the number of cells.

Authors’ response: In the method of Invation assay, quantification method is written.

  1. In the Discussion section, the authors only discuss Figure 4, while the other images are not discussed, and this section needs to be revised.

Authors’ response: Thank you for raising this point. We discussed about other Figures and edited in discussion part.

  1. The authors have only performed cellular experiments, and it would be better to perform in vivo experiments if available.

Authors’ response: Thank you for this great suggestion. We have plans to perform in vivo studies using by mouse model in the feature.

Round 2

Reviewer 2 Report

Comments and Suggestions for Authors

accept